# Functional Deficits of 5×FAD Neural Stem Cells Are Ameliorated by Glutathione Peroxidase 4

**DOI:** 10.3390/cells11111770

**Published:** 2022-05-28

**Authors:** Nawab John Dar, Ren Na, Qitao Ran

**Affiliations:** 1Department of Cell Systems & Anatomy, University of Texas Health San Antonio, San Antonio, TX 78229, USA; nawabdar@gmail.com (N.J.D.); na@uthscsa.edu (R.N.); 2The Biggs Institute for Alzheimer’s and Neurodegenerative Diseases, University of Texas Health San Antonio, San Antonio, TX 78229, USA; 3Research Service, South Texas Veterans Health Care System, San Antonio, TX 78229, USA

**Keywords:** neural stem cells (NSCs), glutathione peroxidase 4 (Gpx4), neuronal differentiation, lipid peroxidation, Alzheimer’s disease (AD) and 5×FAD

## Abstract

Alzheimer’s disease (AD) is the most common cause of dementia affecting millions of people around the globe. Impaired neurogenesis is reported in AD as well as in AD animal models, although the underlying mechanism remains unclear. Elevated lipid peroxidation products are well-documented in AD. In current study, the role of lipid peroxidation on neural stem cell (NSCs) function is tested. Neural stem cells (NSCs) from 5×FAD mice, a widely used AD model with impaired neurogenesis, were observed to have increased levels of lipid reactive oxygen species compared to NSCs from control WT mice. 5×FAD NSCs exhibited altered differentiation potential as revealed by their propensity to differentiate into astrocytic lineage instead of neuronal lineage compared to WT NSCs. In addition, 5×FAD NSCs showed a reduced level of Gpx4, a key enzyme in reducing hydroperoxides in membrane lipids, and this reduction appeared to be caused by enhanced autophagy-lysosomal degradation of Gpx4 protein. To test if increasing Gpx4 could restore differentiation potential, NSCs from 5×FAD and Gpx4 double transgenic mice, i.e., 5×FAD/GPX4 mice were studied. Remarkably, upon differentiation, neuronal linage cells increased significantly in 5×FAD/GPX4 cultures compared to 5×FAD cultures. Taken together, the findings suggest that deficiency of lipid peroxidation defense contributes to functional decline of NSCs in AD.

## 1. Introduction

Alzheimer’s disease (AD) is the most common cause of dementia, accounting for an estimated 60% to 80% of cases. More than 6 million Americans are living with Alzheimer’s disease. By 2050, this number is projected to rise to nearly 13 million [1]. Reports have shown that neurogenesis occurs in adult human brain and is found to be impaired in AD patients contributing conceivably towards memory decline [2,3]. For example, cognitive decline in Alzheimer’s disease patients was reported to correlate with decreased neurogenesis when compared with non-demented individuals with Alzheimer’s like pathology [4]. In addition, familial Alzheimer’s disease patients were revealed to have altered hippocampal neurogenesis (AHN) in postmortem brains [5,6,7]. Moreover, AHN impairment in dentate gyrus was shown to appear much earlier than pathological markers (senile plaques and neurofibrillary tangles) in early stages of AD [8]. Studies using AD mouse models have also documented that impaired neurogenesis is correlated with AD hallmarks and that this impairment manifests as proliferation deficits, reduced neuronal differentiation, increased gliogenesis or differentiation into glial lineage than neuronal phenotype [9,10,11]. Nevertheless, there is no clear understanding of the factors contributing towards impaired neurogenesis. 

Oxidative stress is often a result of redox imbalance caused by alterations in metabolism and antioxidant defense which lead to increased oxidative damage [12]. Accumulating evidence suggest that oxidative stress plays a crucial role in the initiation and progression of AD pathogenesis [13]. Among the antioxidant systems, glutathione peroxidases (GPXs) are evolutionarily conserved enzymes that use glutathione (GSH) as cofactor to reduce peroxides (e.g., R–OOH) to their corresponding alcohols (e.g., R–OH), thereby limiting the transition metal-dependent formation of free radicals (e.g., R–O^•^) [14]. Notably, a high load of phospholipid hydroperoxides (PLOOHs) induces ferroptosis, an iron-dependent oxidative mode of cell death that is hallmarked distinctively from other modes of cell death such as apoptosis [15]. Gpx4 is regarded as the master regulator of ferroptotic cell death because of its ability to reduce PLOOHs to their corresponding non-toxic phospholipid hydroxides (PLOHs), which limits damage to the membranes and dissemination of lipid peroxidation chain reaction [16,17]. Neurogenesis is shown to be sensitive to various types of stress, in particular to oxidative stress [18]. Alterations in neurogenesis and stem cells functions have been associated with increased lipid peroxidation in aging as well as Alzheimer’s disease [19]; however, it is unclear how increased lipid peroxidation and associated oxidative stress may affect the function of Neural Stem Cells (NSCs), which are the most primordial and uncommitted cells of the nervous system that proliferate without limit to produce progeny cells which terminally differentiate into neurons, astrocytes and oligodendrocytes. NSCs are responsible for generating new neurons in adult brain that reportedly integrate into functional neurons and have shown to ameliorate cognitive impairments in AD models [20,21] and this functional integration depends largely on their potential to differentiate into specific lineage [22,23]. Because of their ability of terminal differentiation into specific lineage (neuronal or glial), NSCs are considered to have therapeutic potential in neurodegenerative diseases. In present study, we assessed the differentiation function of NSCs from 5xFAD mice, a widely used AD mouse model [24], and tested the effect of enhanced defense against lipid peroxidation through Gpx4 overexpression on differentiation potential. Our findings suggest that enhanced protection against lipid peroxidation is necessary for maintaining neuronal differentiation ability of NSCs. 

## 2. Materials and Methods

### 2.1. Reagents

Dulbecco’s Modified Eagle Medium: Nutrient Mixture F-12 1:1 containing glucose (Corning, Manassas, VA, USA), HEPES, glutamine, basic fibroblast growth factor, heparin, recombinant human epidermal growth factor, antibiotic/antimycotic, transferrin, progesterone, insulin, putrescine, sodium selenite, All trans-retinoic acid, Penicillin G sodium salt, Streptomycin sulfate, Phosphate buffered saline, RIPA buffer, Accutase, anti-GAPDH antibody (#G9545) (Sigma-Aldrich Inc. Saint Louis, MO, USA), protease inhibitors (EMD Biosciences Inc., San Diego, CA, USA), PierceTM Bicinchoninic acid (BCA) assay kit, BSA, Pre-stained protein ladder (Thermo scientific, Waltham, MA, USA), Fetal bovine serum (Hyclone), antibodies against Tuj1(#sc-80005), GFAP (#ZRB2383), SNAP-25 (#sc-20038), GPX4 (#sc-166570) and (#sc-50497), *β*-actin (#sc-47778) (Santa Cruz Biotechnology, Dallas, TX, USA), Alexa Flour 488 goat anti-rabbit (#A11008), Alexa Fluor 594 goat anti-mouse (#A11005) from (Invitrogen, Waltham, MA, USA), anti-Sox-2 (#AF 2018) from (R&D systems, Minneapolis, MN, USA), Nestin (#NES) from (Aveslabs, Davis, CA, USA) Anti-APP C-terminal antibody (#A8717) from EMD Millipore (Billerica, MA, USA), anti- NeuN (#24307), anti-rabbit HRP linked (#7074), and anti-mouse HRP-linked antibody (#7076) from (Cell Signaling Technology, Beverly, MA, USA), AmershamTM ECL Western Chemilumenescent HRP substrate (Global Life sciences Solution, Buckinghamshire, UK), PVDF membrane from Merck Millipore (Darmstadt, Germany).

### 2.2. Isolation of Neural Stem cells NSCs

5×FAD mice were purchased from Jackson Labs (in B6 background, #34848-JAX). Tg (GPX4) mice (in B6 background) were maintained in-house. Tg (GPX4) mouse was generated with an endogenous human GPX4 transgene. A human P1 clone containing the intact human GPX4 gene (about 3 kb) plus about 30 and 20 kb of 5′- and 3′-flanking sequence, respectively, was used to generate the transgenic mice [25]. Neural stem cells (NSCs) were isolated from the forebrains of embryonic day (E14.5-E15) mice pups obtained after crossing male 5×FAD mice with female WT or Tg(GPX4) mice and cultured as described previously [26,27]. Briefly, the pups were decapitated, and forebrains were dissected. The dissected forebrain tissues were homogenized in NSC media (Dulbecco’s Modified Eagle Medium: Nutrient Mixture F-12 1:1 containing glucose, HEPES, glutamine, basic fibroblast growth factor (bFGF;Upstate, 20 ng/mL), heparin (Sigma, 2μg/mL) recombinant human epidermal growth factor (rhEGF; Sigma, 20 ng/mL) antibiotic/antimycotic and hormone mix [DMEM:F12, glucose (0.6%), transferrin (1 mg/mL), progesterone (0.2 μM), insulin (0.25 mg/mL), putrescine (0.097 mg/mL) and sodium selenite (0.3μM)] and plated at a density of 10^5^cells/cm^2^ in serum-free full NSC media. Cells from two or three pups were pooled together based on our genotyping results and was considered as one group. Using this strategy each experiment was repeated three times. Cells were cultured for 7 days to generate neurospheres and these neurospheres were dissociated using accutase. The dissociated neurospheres were plated as monolayers in desired culture plates or chamber dishes coated with Poly-D-Lysine (Sigma, St. Louis, MO, USA) for further experiments. Procedures for handling mice in this study were reviewed and approved by the Institutional Animal Care and Use Committees of the University of Texas Health San Antonio and Audie Murphy Memorial Veterans Hospital, South Texas Veterans Health Care System. All methods were performed in accordance with the relevant guidelines and regulations.

### 2.3. Lipid Reactive Oxygen Species (ROS) Assay via Liperfluo Staining

Basal lipid ROS levels in neural stem cells (NSCs) were measured using Liperfluo dye according to the manufacturer instructions (Dojindo Molecular Technologies, Inc, Rockville, MD USA). Briefly, the neurospheres were dissociated to single cells by Accutase (Sigma-Aldrich Inc., St. Louis, MO USA) treatment and seeded on Poly-D-Lysine coated plates at a density of 10^5^/cm^2^. The cells were incubated in a humidified atmosphere of 95% air and 5% CO_2_ at 37 °C and fed with NSC media till they reached proper confluency. Finally, these cells were incubated at 37 °C for 30 min with 10uM final concentration of Liperfluo dye (dissolved in DMSO) and images were taken using EVOS M7000 Imaging System (Invitrogen, Carlsbad CA, USA) coupled with EVOS onstage incubator for precise control of temperature, CO_2_, and humidity. Images were taken at 20× objective using a GFP LED cube. Quantification of Liperfluo intensity was done by multiplying fluorescence intensity mean X% area from each group via ImageJ 1.52a (NIH, USA) and represented as relative fluorescence units (arbitrary units). For flow cytometric analysis, the cells were incubated with 10uM final concentration of Liperfluo dye at 37 °C for 30 min and washed with 1×PBS. Finally, the cells were collected in PBS transferred to flow acquisition tubes and quantified using BDLSR II (BD Biosciences, San Jose, CA, USA).

### 2.4. Protein Expression by Immunoblotting

Neural stem cells (NSCs) grown as neurospheres as well as monolayers were lysed in RIPA buffer (20 mM Tris, pH 7.4, 0.25 M NaCl, 1 mM EDTA, 0.5% NP-40, and 50 mM sodium fluoride) supplemented with protease and phosphatase inhibitors (EMD Biosciences Inc., San Diego, CA, USA) for 30 min on ice as described previously [28]. The cell suspension was centrifuged at 14,000 rpm for 15 min at 4 °C, and the supernatant was collected and stored at −20 °C for future use. Protein estimation was performed using a BCA kit (Thermo scientific, MA, USA) according to manufacturer instructions. 30 µg total protein per sample was separated by SDS-PAGE and transferred to PVDF membranes. Membranes were blocked with 5% BSA and then incubated with primary antibody overnight at 4°C. After incubation the membranes were washed with 1×TBST (pH 7.7) thrice for 5 min each and incubated with horse reddish peroxidise (HRP) conjugated anti-rabbit or anti-mouse secondary antibodies (Cell Signaling Technology, Beverly, MA, USA) for 1h at room temperature. Following incubation, the membrane was washed three times with 1×TBST (pH 7.7) for 15 min each. The bands were visualized using an ECL Kit (GE Healthcare, Piscataway, NJ, USA) on Amersham Imager 600 system. The bands were quantified using NIH ImageJ software (ImageJ 1.52a; http://imagej.nih.gov/ij access on 23 May 2022) and normalized to the loading control (*β*-Actin or GAPDH).

#### Autophagy-Lysosomal degradation of GPX4

Gpx4 levels were measured via immunoblotting described above after incubating with autophagy inhibitors. Briefly the NSCs were treated with Bortezomib (100nM), a proteasome inhibitor for 6 h and with Baflomycin A1(Baf.A1) 100nM, a chemical inhibitor of autophagy-lysosomal pathway for 6 h and compared Gpx4 protein levels by immunoblotting.

### 2.5. Real-Time qRT-PCR 

Total RNA was isolated with Trizol reagent (Sigma-Aldrich, St. Louis, MO, USA) and then reverse transcribed using the iScript RT kit (Bio-Rad, Hercules, CA, USA). Real-Time qRT-PCR was conducted as described previously [29]. The mRNA levels were normalized to *β*-Actin to control for input RNA. Primers used were as follows: Gpx4 (forward: 5′-AGT ACA GGG GTT TCG TGT GC-3′; reverse: 5′-CAT GCA GAT CGA CTA GCT GAG -3′); *β*-Actin (forward: 5′-ATC TGG CAC CAC ACC TTC TAC-3′; reverse: 5′-CAG GTC CAG ACG CAG GAT G-3′).

### 2.6. Cell Proliferation Assay

Cell proliferation potential between the WT and 5×FAD NSCs was investigated using Click-&-Go Edu 647 Flow cytometry kit (Click Chemistry Tools, Scottsdale, AZ, USA) according to the manufacturer instructions. Briefly the NSCs were incubated with 10uM of Edu for 6 h and washed with wash buffer containing 1% BSA in PBS. The cell pellet was collected via centrifugation, fixed and permeabilized. A reaction cocktail containing PBS, copper catalyst, fluorescent azide and reducing agent was added and incubated for 30 min in dark area. Following incubation, the cells were washed with 1x saponin-based wash buffer and proceeded for detection using BD FACS Calibur™ (BD Biosciences, San Jose, CA, USA) flow cytometer.

### 2.7. Differentiation of Neural Stem Cells (NSCs)

Neural stem cells were dissociated using accutase and plated on chamber glass slides for immunofluorescence imaging. The differentiation potential was investigated as described previously [30], using known differentiation agent, retinoic acid (RA). Briefly, the NSCs were grown as monolayers in NSC feeding media with 10uM final concentration of retinoic acid. The cells were fed every alternating day with conditioning media and observed continuously for neurite outgrowth under phase contrast microscope. On day 7, the cells were washed with 1×PBS and fixed with 4% Paraformaldehyde (PFA) at room temperature for 15 min. After fixation the cells were washed again with 1×PBS and processed for Immunocytochemistry. 

### 2.8. Immunofluorescence Staining

Immunofluorescence studies were carried out as reported previously [31]. Briefly, NSCs were grown in Poly-D-Lysine coated chamber glass slides and differentiated using retinoic acid for 7 days. On day 8, NSCs cultures were washed twice with PBS and covered up to a depth of 2–3 mM with 4 % formaldehyde diluted in warm PBS and allowed to fix for 15 min at room temperature. After fixation, the cells were rinsed 5 times with 1× PBS and permeabilized with triton X-100 (0.5 %) for 10 min and washed with 1× PBS three times for 5 min each and blocked in blocking buffer (5% normal goat serum in PBS) for 60 min. The blocking buffer was aspirated, and cells were incubated with primary antibody (Tuj1 and GFAP) 1:100 dilution each in antibody dilution buffer (1% BSA in PBS) overnight at 4 °C. After overnight incubation, cells were washed 5 times with 1× PBS for 5 min each and incubated with fluorochrome conjugated secondary antibody (1:500) for 1 h at room temperature. Then, cells were washed with 1× PBS in the dark three times for 5 min each, and finally, 1 μL/mL DAPI (1 mg/mL stock) was added and incubated for 5 min and washed again three times with 1xPBS. After final washing the images were captured on EVOS M7000 Imaging System (Invitrogen, Carlsbad CA, USA) by using DAPI, GFP and RFP LED cubes. Image analysis for fluorescence intensity of each cell population (% of immunopositive cells for each marker) was done via manual scoring for positive or negative for lineage markers in regularly spaced fields in each image via ImageJ 1.52v (NIH). The percentage of positive cells against each lineage specific marker was determined out of the total number of counted cells stained by DAPI.

### 2.9. Statistical Analysis

Numerical data were presented as mean ± standard deviation (SD) of three independent experiments. The statistical analysis by ANOVA indicated significant overall difference between the groups and Students’ *t* test revealed significant pair wise difference by using GraphPad Prism 8.0.1 (244) (GraphPad Software, San Diego, CA, USA). Statistical significance was set to a minimum of *p* < 0.05.

## 3. Results

### 3.1. 5×FAD NSCs Exhibit Elevated Lipid ROS and Reduced Neuronal Differentiation

Increased lipid peroxidation products are readily detected in AD brains and in several AD mouse models [32,33,34] and has shown to affect NSC behaviors [19,35]. 5×FAD mice, a widely used AD mouse model which expresses human APP and PSEN1 transgenes with three (3) of the AD-linked mutations in APP gene [36], are reported to exhibit impaired neurogenesis [37]. This prompted an investigation into whether NSCs from 5×FAD mice might show altered lipid peroxidation. As described in methods, NSCs were isolated from WT and 5×FAD mice. NSCs isolated on day 14 (E14) were cultured for 7 days to generate neurospheres and passaged further to generate secondary neurospheres and these secondary neurospheres were dissociated to grow as monolayers for differentiation experiments on Poly-D Lysine coated surfaces. To characterize these NSC before and after differentiation, they were probed with Sox-2 and Nestin in both neurospheres as well as monolayer conditions. We probed cells from both wildtype animals as well as from 5×FADs in order to establish whether the isolation and culturing of NSCs is comparable between genotypes and found that there was no difference in the marker expression along the groups. As shown in Appendix A, the neurospheres were positive for both Sox-2 and nestin (specific NSC markers) and with differentiation the expression was decreased in monolayers (Appendix A) indicating the differentiation efficiency in these cells.

5×FAD NSCs showed normal proliferation/growth rate (Appendix A) and as expected, had increased APP levels (Figure 1B). To assess lipid peroxidation, we analyzed the lipid reactive oxygen species (ROS) levels by florescence microscopy as well as by flow cytometry using Liperfluo dye. Liperfluo specifically detects lipid ROS. Its oxidized form is nearly nonfluorescent in an aqueous media but emits a strong fluorescence in lipophilic sites such as in cell membranes [38]. Interestingly, increased fluorescence intensity of Liperfluo oxidation was observed in 5×FAD NSCs cells compared to WT cells (Figure 1C) indicating increased lipid ROS in 5xFAD group. Similarly, flow cytometry analysis indicated an increased Liperfluo florescence intensity in 5xFAD cells compared to WT (Figure 1D). One of the key characteristics of NSCs is their ability to differentiate precisely into specific lineage which affects functions such as cognition [39,40]. Because increased lipid peroxidation is associated with cognition deficits, we were interested in assessing differentiation potential of 5×FAD NSCs into specific lineages. NSCs from 5×FAD and WT mice were disassociated into monolayer cells and differentiated using a well-established retinoic acid protocol. The differentiated cultures were then probed for the expression of neuron- and astrocyte- specific markers by immunoblotting. Tuj1, a neural specific protein, is a widely regarded neuronal marker whereas GFAP is used as an astrocyte-specific marker in differentiated cultures [41,42]. Interestingly, Tuj1 levels were significantly higher in WT cultures compared to 5×FAD cultures (Figure 1E,F) whereas GFAP level was higher in 5×FAD cultures compared to WT cultures (Figure 1E,F). This result suggests that 5×FAD NSCs had altered differentiation potential with a deficit of differentiation ability into neuronal lineage. Consistently with the deficit of ability to differentiate into neuronal lineage, reduced level of synaptosomal-associated protein-25 (SNAP-25), a synaptic protein expressed in differentiated mature neurons [43] [44], was observed in 5×FAD cultures compared to WT cultures (Figure 1E,F).

### 3.2. Reduced Gpx4 in 5×FAD NSCs

Because Gpx4 has a key role in defense against lipid peroxidation, the increased lipid ROS in 5×FAD NSCs prompted us to inspect the status of Gpx4 in 5×FAD NSCs. NSCs from 5×FAD and WT mice were cultured as neurospheres as well as monolayers, and Gpx4 levels were analyzed by immunoblotting. Notably, we found a significantly decreased level of Gpx4 protein in 5×FAD NSCs compared to WT NSCs in both neurospheres (Figure 2A) and monolayers (Figure 2B). Thus, the reduced Gpx4 could be a mechanism contributing to the increased lipid peroxidation and functional deficits in 5×FAD NSCs. To determine how Gpx4 deficiency may have occurred, we decided to analyze whether the changes are happening at the transcriptional or at post transcriptional levels. We measured Gpx4 mRNA levels by RT-qPCR and found no significant difference in Gpx4 mRNA levels between WT and 5×FAD NSCs (Figure 2C) indicating that the decrease in Gpx4 protein level occurred post-transcriptionally.

### 3.3. Autophagy-Lysosomal Degradation of GPX4 Was Responsible for Gpx4 Deficit of 5×FAD NSCs

Protein levels within cells are determined not only by rate of synthesis, but also by rates of degradation. The finding of decreased Gpx4 protein yet without reduction of Gpx4 mRNA prompted us to determine whether there is altered degradation of Gpx4 protein in 5×FAD NSCs. The ubiquitin proteasomal pathway and the autophagy-lysosomal pathway are responsible for degrading proteins and organelles. To understand the mechanism by which Gpx4 is degraded in 5×FAD NSCs, we tested whether Gpx4 protein level could be restored by inhibiting the ubiquitin proteasomal pathway or the autophagy-lysosomal pathway using chemical inhibitors. We first treated the NSCs with Bortezomib, a proteasome inhibitor, and found no significant change in Gpx4 protein between treated and untreated groups (Figure 3A,B), indicating that Gpx4 is not being degraded in 5×FAD NSCs by the proteasomal pathway. We next treated the cells with Baflomycin A1(Baf.A1), a chemical inhibitor of autophagy-lysosomal pathway, and compared Gpx4 protein by immunoblotting. Significantly, Gpx4 protein level in Baf.A1-treated 5×FAD NSCs was similar to that of Baf.A1-treated WT NSCs (Figure 3C), indicating that Gpx4 deficit in 5×FAD NSCs is a result of augmented degradation of Gpx4 protein through the autophagy-lysosomal pathway.

### 3.4. Overexpression of Gpx4 Restored Differentiation Function of 5×FAD NSCs

Since we found decreased levels of Gpx4 in 5×FAD NSCs and impaired differentiation potential compared to WT NSCs, we were interested in determining if overexpressing Gpx4 could attenuate differentiation deficits of 5×FAD NSCs. Tg (GPX4) mouse is a transgenic mouse model generated using a human endogenous GPX4 gene [25]. We mated Tg (GPX4) mice with 5×FAD mice, and obtained NSCs from double transgenic 5× FAD/GPX4 pups.

NSCs from Tg(GPX4) were also obtained and designated as GPX4 OE for Gpx4 overexpression. To confirm the Gpx4 overexpression in 5×FAD/GPX4 NSCs, we performed immunoblotting to compare Gpx4 protein levels between 5×FAD/GPX4 and control 5×FAD NSCs. As shown in Figure 4A,B, there was a significant increase in Gpx4 protein in 5×FAD/GPX4 NSCs compared to control 5×FAD NSCs. Next, we were interested in investigating whether overexpression of GPX4 could suppress the increased lipid peroxidation levels in 5×FAD NSCs. We compared lipid ROS levels via Liperfluo oxidation among NSCs of 5×FAD/GPX4, 5×FAD, GPX4 OE and WT genotypes and found that lipid ROS levels were attenuated in 5×FAD/GPX4 and GPX4 OE NSCs compared to control 5×FAD NSCs (Figure 4C,D).

We next assessed differential potential of 5×FAD/GPX4 NSCs. 5×FAD and 5×FAD/GPX4 NSCs were differentiated by the retinoic acid protocol. As controls, WT and GPX4 OE NSCs were also differentiated using the same protocol. To assess the numbers of neurons, we compared levels of neuronal marker Tuj1 protein among the differentiated cultures. As shown in Figure 4E,F, the highest level of Tuj1 was observed in GPX4 OE culture. And, compared with 5×FAD culture, 5×FAD/GPX4 culture showed significantly higher level of Tuj1. To confirm the immunoblotting results, we also co-stained the differentiated cultures and observed by immunofluorescence for astrocytic marker GFAP protein and neuronal marker Tuj1 and NeuN proteins. Majority of Tuj1 and NeuN positive cells were observed in differentiated WT and GPX4 OE NSCs cultures compared to differentiated 5×FAD NSCs culture (Figure 5A,C). Notably, more Tuj1 positive cells were observed in 5×FAD/GPX4 NSCs culture compared to 5×FAD culture. We further measured and compared fluorescence intensity of Tuj1, NeuN and GFAP among the cultures. As shown in Figure 5B,D, higher intensity level of Tuj1 fluorescence was observed in 5×FAD/GPX4 culture compared to 5×FAD culture. On the contrary, GFAP fluorescence intensity level was higher in 5×FAD culture than 5×FAD/GPX4 culture (Figure 5B,D). The results thus indicate that increased Gpx4 level promotes differentiation of neuronal lineage in 5×FAD NSCs.

## 4. Discussion

In current study, we showed that 5×FAD NSCs have increased levels of lipid ROS compared to WT NSCs and that 5×FAD NSCs exhibited altered differentiation potential with reduced ability to differentiate toward neuronal lineage. We also showed that 5×FAD NSCs have a deficiency of Gpx4. We further showed that decreased potential of neuronal lineage differentiation was restored in 5×FAD/GPX4 NSCs. The findings of this study show for the first time that there is a direct link between reduced Gpx4 levels, increased lipid ROS and reduced stem cell function from 5×FAD mice. Impaired neurogenesis has been found as an early deficit in the etiology of familial Alzheimer’s disease in transgenic mice [45], and NSCs from AD mice are reported to fail to terminally differentiate into mature neurons [37]. Our findings are thus consistent with those reports indicating that there are functional deficits of NSCs in AD which may contribute to impaired cognitive function. The role of lipid peroxidation in early AD has received special attention in recent years to improve our understanding in deciphering the pathophysiologic processes involved in AD [46]. Consistent with reports by others [47,48], our report indicates that lipid peroxidation can lead to functional deficit of stem cells. At present, the molecular basis of lipid peroxidation-induced functional deficits of stem cells remains unclear and warrants further investigation in future studies.

Gpx4 is regarded as the master regulator of ferroptosis, a relatively new form of iron dependent cell death which results due to lethal accumulation of phospholipid hydroperoxides in membranes. We have previously shown that conditional knockout of Gpx4 in neurons induced neuronal death in adult animals that was consistent with ferroptosis [29,49]. The findings of this study indicate that Gpx4 also regulates neural stem cell function. Increased oxidative stress and imbalanced redox state in neurogenic microenvironment has been associated to reduced neurogenesis [50]. A recent report indicated that Gpx4 contributes to maintaining the stemness phenotype of cancer stem-like cells [51]. Thus, Gpx4 appears to be a regulator of stem cell function as well. However, the significance of this novel function of Gpx4 for health and disease remains to be established. In this study, we further showed that the deficiency of Gpx4 in 5×FAD NSCs occurred at post transcriptional level, likely through enhanced degradation of Gpx4 protein. Gpx4 is reported to be degraded through chaperone-mediated autophagy (CMA) [52]. Altered functions of autophagy-lysosomal degradation pathway are well-documented in AD [53,54]. Consistent with those findings, our results indicate that the deficiency of Gpx4 in 5×FAD NSCs is likely due to an exacerbated degradation through the autophagy-lysosomal degradation pathway. At present, the precise mechanism responsible for enhanced autophagy-lysosomal degradation of Gpx4 in 5×FAD NSCs is unclear. Future studies will be needed to illustrate the mechanism(s) that specifically target Gpx4 in these cells.

Neural stem cell functions of self-renewal and precise differentiation into neuronal phenotype hold an immense promise from NSC therapeutics perspective. Reports have shown that the improvements in cognitive functions due to treatment with NSCs depend primarily on the precise differentiation into specific lineage [39,40]. In this study, we compared differential potential of WT and 5×FAD NSCs with or without Gpx4 overexpression. Interestingly, overexpression of Gpx4 leads to increased neuronal in 5×FAD background. This finding suggests that enhanced defense against lipid peroxidation is essential for maintaining stem cell function in therapeutic development. The finding also suggests that enhancing Gpx4 activity may be an effective approach to improve NSCs function in therapeutic development. In summary, the findings of this study indicate that there is a direct link between increased lipid ROS and reduced stem cell function. The findings also reveal that Gpx4 plays a key role in maintaining neuronal differential potential of NSCs.

## 5. Limitation of the Study

While this study is an important and appropriate model for studying mechanisms of NSCs differentiation, the mechanisms of adult neurogenesis, especially in the context of AD, may be different and cannot be generalized to adult neurogenesis in vivo. Thus, further studies are warranted to establish the effect of Gpx4 overexpression in increased neurogenesis.

## Figures and Tables

**Figure 1 cells-11-01770-f001:**
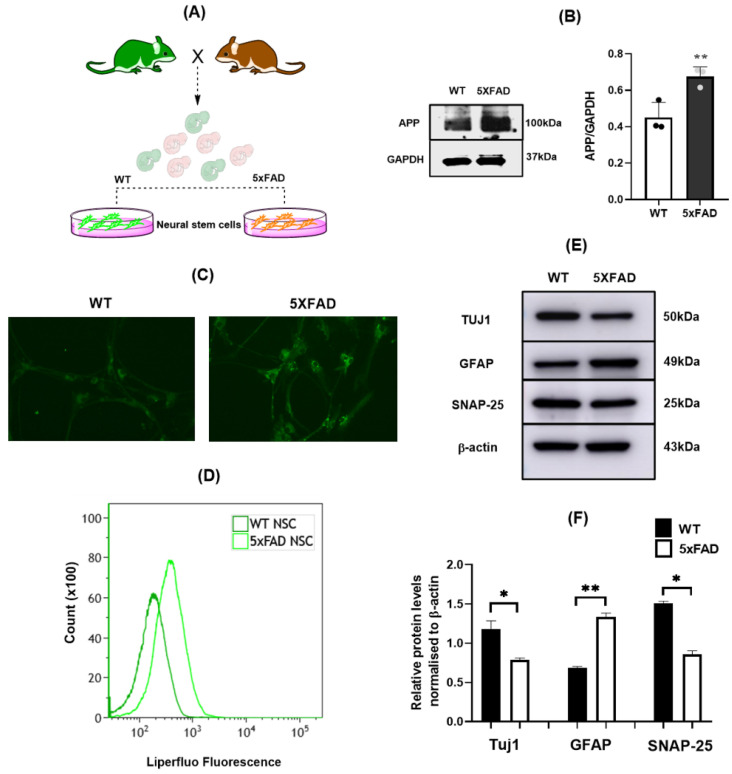
Increased lipid ROS and differentiation deficits in 5×FAD NSC’s: (**A**). Graphical representation of culturing NSC’s from 5×FAD and WT. (**B**). Western blots and densitometric analysis showing increased APP levels in 5×FAD NSC’s. (**C**). Florescence microscopy images representing increased lipid ROS levels in 5×FAD cells compared to WT via fluorescence intensity of Liperfluo dye. (**D**). Flow cytometry quantification of lipid ROS showing higher fluorescence of Liperfluo in 5×FAD cells compared to WT cells. (**E**). Western blot representing expression of Tuj1, GFAP and SNAP-25 in 5×FAD and WT NSC’s. (**F**). Densitometric analysis indicating levels of Tuj1, GFAP and SNAP in differentiated WT and 5×FAD cultures. *: *p* < 0.05; **: *p* < 0.01.

**Figure 2 cells-11-01770-f002:**
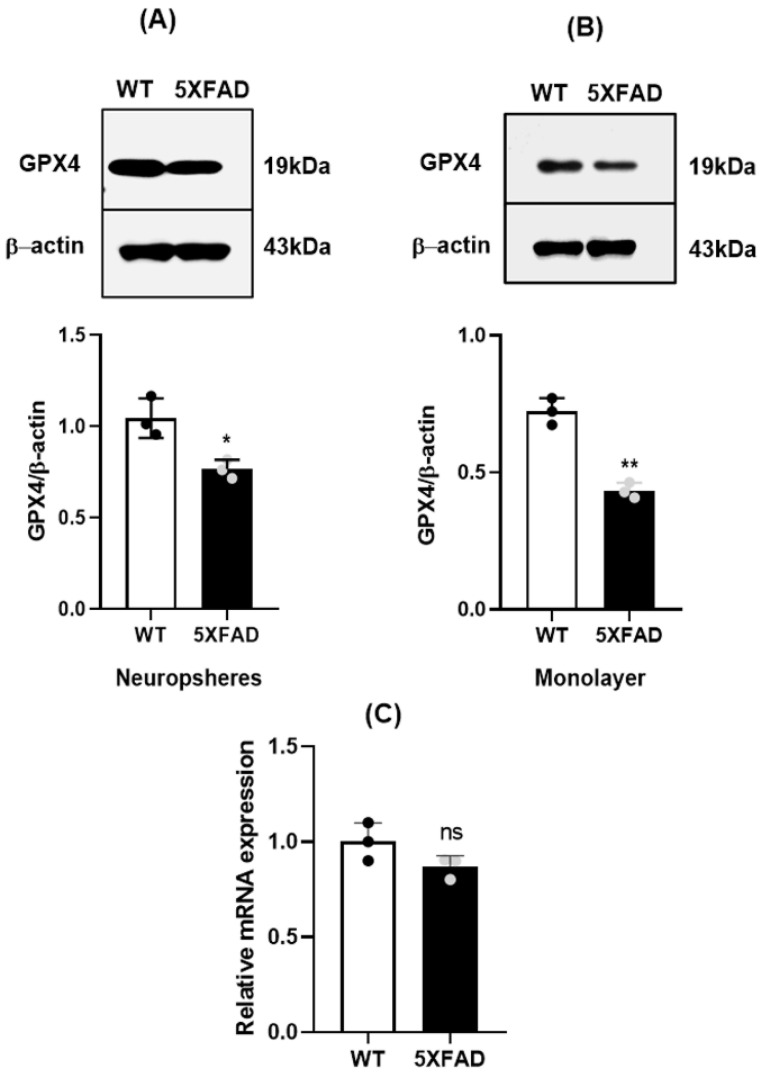
Decreased Gpx4 levels in in 5×FAD NSCs (**A**). Western blot and densitometric analysis indicating decreased levels of Gpx4 in 5×FAD NSCs grown as neurospheres. *: *p* <0.05. (**B**). Western blot and densitometric analysis indicate the decreased levels of Gpx4 in 5×FAD NSCs grown as monolayers. **: *p* <0.01. (**C**). Relative mRNA levels of Gpx4 in 5×FAD and WT NSCs. ns: not significant.

**Figure 3 cells-11-01770-f003:**
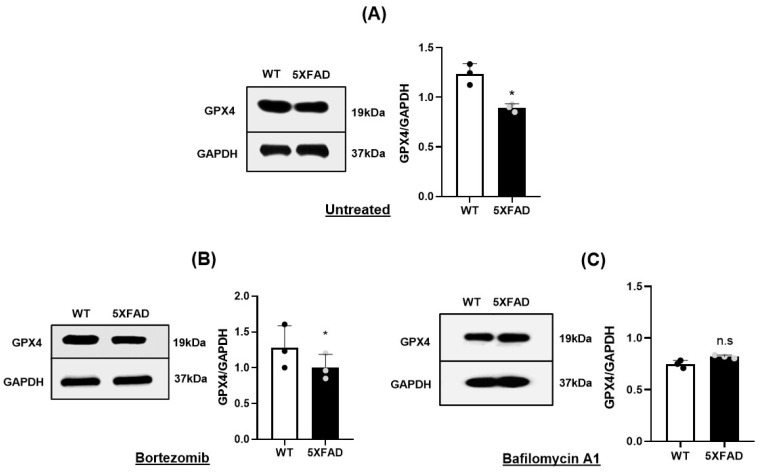
Inhibition of autophagy-lysosomal pathway restores Gpx4 level in 5×FAD NSCs: (**A**). Western blot and densitometric analysis indicate the decreased levels of Gpx4 in 5×FAD NSCs grown as neurospheres. *: *p* < 0.05. (**B**). Western blot and densitometric analysis indicate the decreased levels of Gpx4 in 5×FAD NSCs treated with Bortezomib (a proteasomal inhibitor) compared to WT NSCs treated with Bortezomib. *: *p*< 0.05. (**C**) Western blot and densitometric analysis indicate the levels of Gpx4 in 5×FAD and WT NSCs treated with Bafilomycin A1 (an autophagy inhibitor). ns: not significant.

**Figure 4 cells-11-01770-f004:**
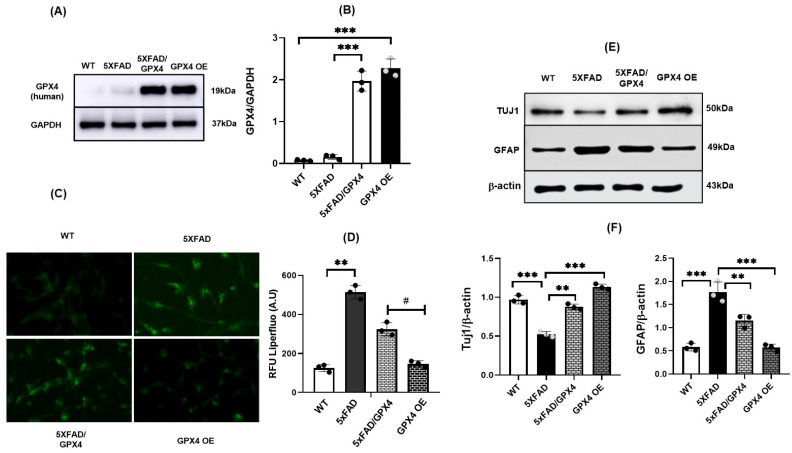
Increased level of neuronal marker protein in differentiated 5×FAD NSCs culture with overexpression of GPX4. (**A**,**B**) Western blot and densitometric analysis indicate the increased levels of human Gpx4 in 5×FAD/GPX4 and GPX4 OE NSCs. *: *p* < 0.05. (**C**) Florescence microscopy images representing increased lipid ROS levels in 5×FAD cells and attenuation of these levels with overexpression of GPX4 measured via fluorescence intensity of Liperfluo dye. (**D**). Quantification of Liperfluo relative florescence intensity (arbitrary units) in WT, 5×FAD, 5×FAD/GPX4 and GPX4 OE NSCs indicating decreased lipid ROS in 5XFAD/GPX4 and GPX4 OE groups compared to 5×FAD only. # represents the significance of reduced lipid ROS in comparison to 5×FAD cells. (**E**). Western blot indicating Tuj1 and GFAP expression in differentiated WT, 5×FAD, 5×FAD/GPX4 and Gpx4 OE cultures. (**F**). Densitometric analysis indicating levels of Tuj1 and GFAP in WT, 5×FAD, 5×FAD/GPX4 and Gpx4 OE differentiated cultures. *: *p* < 0.05. **: *p* < 0.01, ***: *p* < 0.001.

**Figure 5 cells-11-01770-f005:**
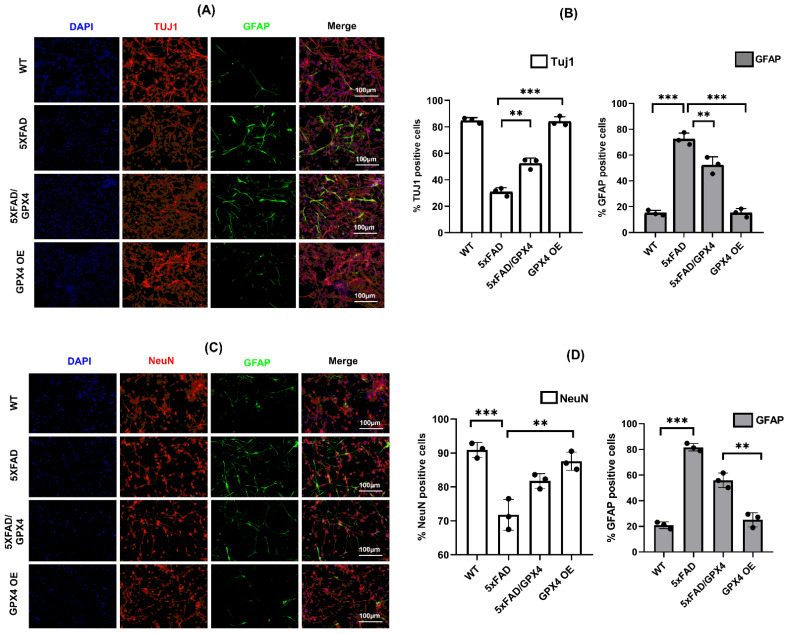
Levels of neuronal and astrocytic markers in differentiated NSCs culture assessed by immunofluorescence (**A**). Immunofluorescent images representing the expression of Tuj1 (Red) and GFAP (Green) in differentiated NSCs cultures. The cultures were also stained with nuclear stain DAPI (blue). (**B**). % of cells positive against Tuj1 and GFAP among differentiated WT, 5×FAD, 5×FAD/GPX4 and GPX4 OE NSCs cultures out of the total number of cells stained with DAPI (**C**). Immunofluorescent images representing the expression of NeuN (Red) and GFAP (Green) in differentiated NSCs cultures. The cultures were also stained with nuclear stain DAPI (blue). (**D**). % of cells positive against NeuN and GFAP among differentiated WT, 5×FAD, 5×FAD/GPX4 and GPX4 OE NSCs cultures out of the total number of cells stained with DAPI. Fluorescence intensity of TUJ1, NeuN and GFAP was measured using Image J software and expressed as % of positive cells against each protein. **: *p* < 0.01, ***: *p* < 0.001.

## Data Availability

The raw data supporting the conclusion of this article will be made available by the authors, without undue reservation.

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
