# Peer review of "Functional Deficits of 5×FAD Neural Stem Cells Are Ameliorated by Glutathione Peroxidase 4"

_cells, 2022, doi:10.3390/cells11111770_

Round 1

Reviewer 1 Report

In the manuscript, Dar and colleagues revealed that neural stem cells established from 5xFAD mice show increased lipid peroxidation due to decreased GPX4, and consequently cell function is affected. On the other hand, there are several unclear points, including the cause of the activation of autophagy, which is responsible for the degradation of GPX4. Overall, this manuscript is well written and could be better with some improvements on a few points that the should be addressed by the authors. Specific comments are below.

Major points.

  1. Indeed, GPX4 is slightly decreased (Figure 2) and lipid peroxidation also occurs (Figure 1). However, cell proliferation is unaffected and ferroptosis has not occurred (Supplementary Figure 2). Therefore I do not understand why the differentiation potential of neural stem cells is altered. The authors claim that the findings of this study show for the first time that there is a direct link between increased lipid ROS and reduced stem cell function (Line 359 to 361). However, the molecular mechanism remains largely unveiled in the manuscript. I suggest that the authors try to provide some evidence, for example, by transcriptome analysis of gene expression including neurogenesis, gliogenesis, neuron fate commitment, synaptic function, or cytokines.

  1. Figure 5: Is it appropriate to compare data by fluorescence? It should be expressed in terms of TUJ1-positive cell percentages or GFAP-positive cell percentages, respectively.

Minor points.

  1. Line 208: 5. xFAD should be 5xFAD.
  2. Line 257: Lipofluo should be Liperfluo.
  3. Figure 3 A to C: I think it would be appropriate to blot these three groups on the same membrane, not separately, for comparison and quantification.
  4. Figure 4: The band of GFAP should be presented as well.
  5. Figure 5: Scale bars should be presented.

Reviewer 2 Report

In the study, the authors investigate the role of Glutathine peroxidase 4 in ameliorating functional deficits in 5XFAD rodent model. There are several discrepancies regarding the manuscript.

How many clones were generated per line? Were multiple clones generated per animal? This information must be included in the manuscript.

More details regarding the Tg (GPX4) mice line must be provided. Or at least the original article must be cited in the method section.

Suppl fig S1. Authors mention decreased expression of Sox-2 and nestin upon differentiation; however, did not provide any evidence for the claim. Was the fluorescence intensities quantified? If so, the information must be provided.

Suppl Fig 1 B: SOX-2 is a nuclear marker, ICC provided here is incorrect.

Fig5: all images are very blur; moreover, NeuN is a nuclear marker and in the figure is shows cytoplasmic. Graphs do not correspond to the ICC data.

Regarding GPX4 antibodies used in the study: The authors mention two antibodies #sc- 166570 and #sc-50497, both raised using recombinant human GPX4 ( aa108-197) and  from Santa Cruz. Surprisingly, it is mentioned in the website that sc-50497 was discontinued and replaced with #sc- 166570. So what antibodies were used in Figure 2, 3 and in particular figure 4, where only human GPX4 is detected. Also based on review both antibodies recognize mouse GPX4 as well.

Line 49: Missing full form for GSH

Figure 1D: Kindly use different colors to differentiate the two peaks (also for other flow-cyto data).

Figure 3: Information about Bortezomib and Baflomycin A1 treatment are missing in the method section and must be provided.

Figure 4E. The images are not sharp. Better representative images must be provided.

Figure 4: Results from characterization of the dual transgenic mice is missing in the results section. At least the authors must show the levels of APP between the two transgenic lines and non-transgenic control.

How many experimental repeats were performed for each experiment? This information regarding no of clones use and experimental repeats must be included.

The authors mention that three independent experiments were performed. Figure 4B and D has only two individual points in the graph? Why? And how did the authors even perform statistical analysis in this case?

Round 2

Reviewer 1 Report

The authors have addressed most of my concerns and I have no further comments.

Author Response

We'd like to thank Reviewer 1 for the thoughtful comments.

Reviewer 2 Report

The manuscript has significantly improved; however, there are still a lot of discrepancies that has to be addressed in the article.

How many clones were generated per line? Were multiple clones generated per animal? This information must be included in the manuscript.

Response: Thank you for your comment, we have updated this information in the revised manuscript under the heading “Isolation of Neural stem cells NSCs” and is highlighted with red.

New comment: Based on the author’s response, only one sample/ group was analyzed for every experiment? If so, the results are not reliable. How many replicates were used for each individual experiment?

Suppl fig S1. Authors mention decreased expression of Sox-2 and nestin upon differentiation; however, did not provide any evidence for the claim. Was the fluorescence intensities quantified? If so, the information must be provided.

Response: Our main goal of doing SOX-2 and Nestin was to characterize the NSC and compare between neurospheres and monolayers. Looking at the florescence intensities of the proteins compared to neurospheres there was a decrease when culturing NSCs as monolayers for differentiation experimets. Quantifying the florescence intensity of one big neurosphere VS the monolayers would not be a fair quantification because of the density and area differences so we did not quantify these images.

New comment: In the figure legend the authors mention “Representative images of monolayers from WT and 5xFAD groups indicating decreased expression of Sox-2 and nestin upon differentiation”. This observation is about monolayers, not spheres vs monolayers. I do not see any reduction in expression of the proteins. Did the authors perform any quantification to claim any decrease in levels of these proteins? If so, results must be provided. Also figure C is missing in the figure legend.

Fig5: all images are very blur; moreover, NeuN is a nuclear marker and in the figure is shows cytoplasmic. Graphs do not correspond to the ICC data.

Response: We have revised our figures and provided with 300 DPI vector images in the revised manuscript. We have presented the quantification data as % cell positivity against each protein. The NeuN protein is localized in nuclei and perinuclear cytoplasm of most of the neurons in the central nervous system of mammals (Acta Naturae. 2015 Apr-Jun; 7(2): 42–47). We have provided an additional cropped image to have a better idea in supplementary figures as Figure S4.

New comment: I do understand that NeuN can also be observed in perinuclear cytoplasm,  but in the provided figure NeuN signal is “only cytoplasmic” (Fig S4), and not neuronal, basically no NeuN protein in nuclei? How?

Regarding GPX4 antibodies used in the study: The authors mention two antibodies #sc- 166570 and #sc-50497, both raised using recombinant human GPX4 ( aa108-197) and from Santa Cruz. Surprisingly, it is mentioned in the website that sc-50497 was discontinued and replaced with #sc- 166570. So what antibodies were used in Figure 2, 3 and in particular figure 4, where only human GPX4 is detected. Also based on review both antibodies recognize mouse GPX4 as well.

Response: Thank you for this comment, since our lab is working on GPX4 for a number of years, we had a good stock of GPX4, H-90 which specifically detects human Gpx4 predominantly, it was recently discontinued and we had discussions with Santa Cruz team about it, but thankfully they came up with E-12 as replacement. E-12 moderately detects mouse GPX4 as well. We have used H-90 in figure 4. Additionally we have developed and purified an antibody against GPX4 in our lab as well which detect mouse Gpx4.

New comment: The mentioned antibody “#sc- 166570”, H-90, also strongly detects mouse GPX4; Ref: Toxicol Lett. 2009; 187(2):69-76; Redox Biol. 2018; 14:535-548; therefore, it is not human specific; the results in Fig 4 are unreliable.

Figure 4: Results from characterization of the dual transgenic mice is missing in the results section. At least the authors must show the levels of APP between the two transgenic lines and non-transgenic control.

Response: APP levels between WT and 5xFAD were presented in Figure 1B, Since we were not interested in looking at the role of GPX4 in attenuating APP levels so we didn’t perform that experiment. In addition, cells from all the groups were genotyped before perfoming lipid peroxidation and differentiation experiments.

New comment: I do appreciate that the authors provided the APP levels of WT and 5xFAD, but the real question is, are there any difference in expression of APP in the dual transgenic model?

The authors mention that three independent experiments were performed. Figure 4B and D has only two individual points in the graph? Why? And how did the authors even perform statistical analysis in this case

Response: Thank you for pointing this out, we have revisited our data and provided new figures in the revised manuscript.

New comment: Surprisingly, the data did not change even after incorporating the new values, how? The figures are the same. Also fig 3B still has only 2 individual values.

Round 3

Reviewer 2 Report

The authors have addressed all the issues raised by this reviewer.